# Influence of Temperature on the Quantity of Bisphenol A in Bottled Drinking Water

**DOI:** 10.3390/ijerph19095710

**Published:** 2022-05-07

**Authors:** Dobrochna Ginter-Kramarczyk, Joanna Zembrzuska, Izabela Kruszelnicka, Anna Zając-Woźnialis, Marianna Ciślak

**Affiliations:** 1Department of Water Supply and Bioeconomy, Faculty of Environmental Engineering and Energy, Poznan University of Technology, Berdychowo 4, 60-965 Poznan, Poland; dobrochna.ginter-kramarczyk@put.poznan.pl (D.G.-K.); marianna.cislak@doctorate.put.poznan.pl (M.C.); 2Faculty of Chemical Technology, Institute of Chemistry and Technical Electrochemistry, Poznan University of Technology, Berdychowo 4, 60-965 Poznan, Poland; joanna.zembrzuska@put.poznan.pl; 3Department of Biophysics, Poznan University of Medical Science, Grunwaldzka 6, 60-780 Poznan, Poland; anna.zajac.woznialis@gmail.com

**Keywords:** bisphenol A, bottled water, drinking water, chromatographic analysis

## Abstract

Bisphenol A (BPA) is a component used in the production of polycarbonate plastics (PC) and epoxy resins, which are currently widely used in food and beverage packaging. Although BPA is not used in polyethylene terephthalate (PET) manufacturing, a recent study reported its presence in PET water bottles. This study was conducted to investigate the effects of storage conditions on the release of BPA from PET bottles as well as to assess health risks associated with the consumption of bottled water. Using high-performance liquid chromatography (HPLC), we measured the content of BPA in local brands of plastic bottled water sold in the Polish market. It has been established that temperature is one of the main factors that influences the migration of bisphenol A to products, as was confirmed by determination of the amount of bisphenol A in water, which was carried out without exposing the bottles to different temperatures. Despite the fact that the individual concentrations of BPA in bottled water were low (ng/L) at 0.6 mg/kg (body weight), the cumulative daily dose in the body may be much higher than the quoted concentrations due to the number of products containing BPA. Thus, prolonged usage of bottled water and beverages should be avoided to reduce the risk of human exposure to BPA through leaching. Additionally, it was found that high temperatures resulted in increased BPA leaching.

## 1. Introduction

Despite numerous restrictions established by the EU [1,2,3,4,5,6,7], there is a growing demand for plastic-based materials, which are still widely used as packaging, e.g., in the food, cosmetics or chemical sectors. This has resulted in increased interest in the compounds used and generated during the production of such packaging. Currently, polyethylene terephthalate (PET) and polycarbonate (PC) bottles are the most common types of water containers. The former is mainly used in the case of lower volumes (ranging from 350 to 5000 mL), especially for the 500 mL bottles that are available in supermarkets and stores. Alternatively, PC is mainly used for the production of packaging with a capacity ranging from 5 to 19 L. Since the use of plastic bottles is widespread, it is also necessary to conduct a comprehensive study regarding the concentration of bisphenol A (BPA) in bottled water (for both PET and PC materials) as well as the associated risk for human health. There are many reports in the literature regarding BPA levels in bottled water (PET), although BPA is not used in the preparation of this material. The possible contamination of polyethylene terephthalate by BPA can occur due to the use of recycled PET (r-PET), which may contain low amounts of BPA originating from printing inks or other materials used in production. Another possible explanation may be associated with the occurrence of cross-contamination, both in the PET recycling process and in the process of obtaining primary PET [4]. Other sources of exposure to BPA may also include bottle caps, the environment, and even BPA-contaminated food products [8,9,10,11,12,13,14,15,16,17,18,19,20]. 

Bisphenol A (BPA, dian, 2,2-di(4′-hydroxyphenyl)propane, diphenylolpropane) is an organic compound that belongs to a group of phenols with the formula C_15_H_16_O_2_ and a molecular weight of 228.28. This compound was first obtained by Aleksander Dianin in 1891, during a reaction between acetone and two phenol molecules (Figure 1). The process is carried out with an excess of phenol in the presence of a catalyst—hydrochloric acid or an ion-exchange resin containing strong sulfonic acid groups (R-SO_3_H). Water is a by-product of this reaction [11,21,22]. 

BPA can undergo various chemical reactions, e.g., hydrogenation, coupling, nitrosation, alkylation, nitration or esterification. In some cases, different final products may be formed, depending on process conditions and the type of reaction. The hydrogenation reaction of BPA, which can be carried out using three different pathways (Figure 2), is an example of this phenomenon. The first hydrogenation pathway results in the formation of phenol and *p*-isopropylphenol, and the second yields cyclohexanol and *p*-isopropylphenol, whereas the third yields 2,2-bis-(4′-hydroxycyclohexyl)propane as well as 2(4′-hydroxycyclohexyl)-2-(4″-hydroxyphenyl)propane [11]. 

For several years, the monitoring of BPA content was particularly focused on surface waters, river sediments, sewage sludge, wastewater, drinking water and food products. These measurements allowed for establishment of the following content ranges: 0.0005 ÷ 0.41 μg/L in surface waters, 0.01 ÷ 0.19 mg/L in river sediments, 0.004 ÷ 1.363 mg/kg in sewage sludge and 0.018 ÷ 0.702 μg/L in wastewater. Recent studies have determined the content of BPA in food products (canned beans, canned vegetables and meat, fresh turkey, infant formulas), and its concentration ranged from 0.23 to 65.0 ng/100 g of analyzed sample [23]. The test results clearly confirmed the presence of bisphenol A in the environment and commercial food products. It is estimated that the high demand and mass production of plastics results in emission of over 100 tons of BPA into the atmosphere each year [11,19,21,22,24].

In consequence, the problems associated with the widespread presence of bisphenol A have become the subject of substantial consideration. Studies regarding the toxic effects of BPA have been carried out on animals (incl. mice, rats, dogs and rabbits), and its effects on the human body were investigated subsequently. Based on the results, it was established that the mechanism of BPA toxicity is based on two phenomena: its irritating properties (BPA is classified as a respiratory irritant and may cause sensitization via skin contact) and its estrogenic activity (BPA belongs to the group endocrine disruptors) [11,22,24,25]. The presence of bisphenol A is detected in urine samples as well as in numerous tissues and organs, blood, fetal serum, human milk and the placenta [25,26]. The effects of BPA on the nervous system result in disturbances in the proper formation of brain sex and behavior [27,28]. It is also suspected that BPA is responsible for schizophrenia, depression and Alzheimer’s disease. Additionally, BPA affects the female reproductive system, which results in the following disorders: increased probability of breast and uterine cancer, polycystic ovary syndrome, premature puberty, problems with becoming pregnant and miscarriages [29]. In 2011, scientists from Sweden confirmed that children born by women exposed to high levels of BPA exhibit low birth weight, reduced immunity, numerous allergies and, in extreme cases, disorders of the development of sexual organs. BPA also adversely affects the male reproductive system. Studies conducted in Japan concerning men working in conditions of particular exposure to the harmful factor confirmed the relationship between increased BPA in urine and a decreased sex drive associated with difficulty in achieving erection, lower ejaculation strength and lower overall satisfaction with sex life. Similar research was conducted by a team of scientists from the University of British Columbia in Canada in 2009 and 2010. Based on the obtained results, it was confirmed that a high level of BPA causes erectile dysfunction and an increase in sexual dysfunction. In extreme cases, long exposure to BPA can lead to male infertility. It should also be emphasized that the development of prostate cancer is one of the most dangerous complications associated with BPA. It has been shown that increased ingestion of BPA is capable of stimulating the proliferation of neoplastic cells. Other disorders related to the presence of BPA in the body include an increased risk of developing diabetes, cardiovascular disease and obesity due to the ability of BPA molecules to activate fat cells (adipocytes) and multiply them excessively. Studies regarding possible routes for the penetration of bisphenol A from packaging into the environment or into the human body have only begun to appear in the last decade. Attempts were carried out to determine the patterns of BPA migration from packaging to products and to establish the factors that influence its migration rate. The likelihood of contamination of a product with bisphenol A increases at elevated temperatures, e.g., when the product is heated, e.g., by pouring hot milk into a bottle or exposing it to direct sunlight. Moreover, the pH (acidic juices or alkaline liquids) is another factor that contributes to BPA migration. The exposure time is also a crucial. Repeated use of the product, e.g., via frequent washing with hot water; scrubbing with sharp steel wool or sponges; or application of other mechanical stresses, such as squeezing, creates conditions favorable for BPA migration. Exposure to its toxic effects occurs when food or drinks contain this harmful substance, through the use of toxin-contaminated cosmetics or other products that readily penetrate the skin. Canada was the first country in the world to officially recognize bisphenol A as a toxic and harmful compound. On 13 October 2010, an amendment to the Environmental Protection Act was published in the Official Journal of Canada, which stated that BPA had been added to the list of toxic substances. Enactment of this new law was a success despite numerous protests from the Canadian and US chemical industries. On 11 March 2010, additional regulations were introduced that prohibited the advertising, sale and import of baby bottles containing BPA admixtures in Canada. After this change in regulations in Canada, European countries also began to restrict the content of this dangerous substance. In Europe, bisphenol A was allowed for production of plastic materials and products intended to come into contact with food under the Commission Directive 2002/72/EC of 2002, which was valid for 9 years. It was not until 14 January 2011, that the new Regulation 10/2011/EU [29] was adopted, which regulated the permissible amounts of bisphenol A in products that come into contact with food and products suitable for human consumption. Starting from 1 March 2011, a total ban on the use of BPA for the production of baby bottles was introduced in the European Union, and, from 1 June 2011, the import of these bottles from countries outside of the European Union was prohibited on the basis of Directive 2011/8/EU. In the EU, the European Food Safety Authority (EFSA) is responsible for collecting and analyzing scientific data and identifying potential threats related to various toxic substances, including BPA. The EFSA conducted several health risk assessments for harmful exposure of the human body to BPA concerning three groups of people: adults, children and infants. On this basis, the EFSA published a scientific opinion and established the Tolerable Daily Intake of TDI at 0.04 mg/kg body weight per day [30,31]. This reduction in the TDI (from 0.05 to 0.04 mg/kg) results from an assessment of studies published in the period between 2013 and 2018, particularly those that indicate the adverse effects of BPA on the immune system. In animal studies, an increase was observed in the number of “T-helper” cells, a type of white blood cell that plays a crucial role in cellular immune mechanisms. At elevated levels, these cells can lead to the development of allergic lung inflammation.

The scientific opinion published by the EFSA did not completely eliminate the dangers associated with BPA. On the contrary, the controversy associated with bisphenol A continues to grow and relevant discussions occur on the global scale. The World Health Organization (WHO) advises against the use of products containing BPA admixtures. 

Since polycarbonate (PC) and polyethylene terephthalate (PET) are widely used as packaging materials for water storage, there is a strong need to investigate the presence and potential risks associated with the presence of estrogenic bisphenol analogues in bottled water. In the case of scientific reports concerning the BPA content in bottled water, it is possible to find results regarding the concentration of this compound depending on the temperature at which the bottled water is stored. However, these data do not give an unambiguous conclusion. There are studies showing an increase in the content of bisphenol A with an increase in storage temperature [32] and contradicting reports that indicate a lack of such correlations [33]. Hence, this study focuses on investigation of the bisphenol A content in bottled water depending on the storage temperature.

The migration of BPA to food products, e.g., water, is determined by performing a direct analysis of the samples. A pre-validated high performance liquid chromatography (HPLC) technique is preferred for the quantitative determination of BPA. The main aim of the present study is to analyze the presence of Bisphenol A in five samples of bottled drinking water purchased from the local market in the city of Poznan in Poland. The study was focused on testing the hypothesis that the concentration of bisphenol A in bottled water depends on the temperature. In our work, we only evaluate the scale of the problem without carrying out a thorough statistical analysis of the obtained results, which we have omitted due to the small number of sampled bottled waters.

## 2. Materials and Methods

A total of five types of water bottled in polyethylene terephthalate (PET) containers from various manufacturers, marked as A, B, C, D and E, respectively, were used in this study. The water bottles were purchased from randomly selected supermarkets in the city of Poznan in Poland. Within each store, the water bottles were also randomly selected from the stocked batches of the product. The production date of these brands was almost identical, and, for each bottle, there was one year of shelf-life before the expiration date would pass. The water bottles were similar in terms of shape and were characterized by the same volume (500 mL). The plastic material was identified as PET based the global identification number visible inside the recycling triangle on the bottom surface of each bottle. Collectively, a total of 25 water bottles were used in this study. The randomly selected water bottles were transported to the Science Research Laboratory of Poznan University of Technology and kept at 4 °C prior to analysis. One series of bottles was analyzed directly after transport to the laboratory. Next, the water in the bottles was exposed to temperatures of 8, 18, 28, 38 and 48 °C for 24 h. Analysis of each sample for the quantitative measurement of BPA was carried out in five repetitions. The selected temperatures were not random; they reflect conditions that can occur at different times of the year in the temperate climate zone (Poland). Extraction of the samples was performed using the solid-state extraction (SPE) method with an Octadecyl C18 column (J. T. Baker). 

A total of 500 mL of the bottled water samples A, B, C, D and E, respectively, were subjected to solid phase extraction. The extraction process consisted of the following steps (Figure 3):

The analytical method based on liquid chromatography coupled with tandem mass spectrometry (LC-MS/MS) was used to determine trace amounts of bisphenol A in the aqueous samples. 

LC analysis was performed using the UltiMate 3000 RSLC chromatographic system from Dionex (Sunnyvale, CA, USA). Five μL samples were injected into a Hypersil Gold C18 RP analytical column (100 × 2.1 mm × 1.9 µm) (Thermo Scientific, Waltham, MA, USA). In the majority of the experiments, the column was kept at 35 °C, and the mobile phase consisted of 5 mM ammonium acetate in water (A) and methanol (B) at a flow rate of 0.2 mL min^−1^. The following gradient was used: 0 min 50% B, 5 min 90% B, 5.5 min 90% B, 6 min 100% B and 6.5 min 100% B.

The LC system was connected to the API 4000 QTRAP triple quadrupole mass spectrometer from AB Sciex (Foster City, CA, USA). BPA was determined by electrospray ionization mass spectrometry (ESI-MS) operated in the negative ion mode. The analyte was detected using the following settings for the ion source and mass spectrometer: curtain gas 10 psi, nebulizer gas 40 psi, auxiliary gas 40 psi, temperature 450 °C, ion spray voltage −4500 V and collision gas set to medium. 

The MS/MS multiple reaction monitoring mode (MRM) parameters used for quantitative BPA determination is shown in Table 1. In the majority of the experiments, deprotonated molecules [M − H]^−^ of phenol were used as precursor ions. 

## 3. Results

The five most common bottled water brands were selected from the products available on the Polish market. In the initial phase of the experiment, the bisphenol A content was analyzed immediately after their purchase. The concentrations of bisphenol were detected at low levels (at ng/L range). Variations can be observed between the samples in consecutive trials. This may result from slight differences between bottles because the water bottles were purchased randomly. The results of the first research stage are presented in Figure 4.

Bisphenol A was detected in all water samples, except for water D. Its highest concentration was determined in water C. The obtained values correspond well with the values found in the literature, ranging from 4.04 ng/L [34] to 4.6 ng/L [35]. There are several aspects that may be the cause of the differences in the content of BPA in the tested samples, e.g., the material of the bottle, the material of the cap, the method of storage at the stages of production and sale, and the materials of the installation used to draw water and fill the bottles.

Afterwards, the content of this compound was determined in the water samples after incubation of the bottles at various temperatures for a period of 24 h.

The obtained results of the bisphenol A concentration in the PET bottled water samples exposed to different temperatures, i.e., 8, 18, 28, 38 and 48 °C, are presented in Figure 5. In each case, the amount of migrated BPA after exposure of the bottle to the appropriate temperature caused changes in its concentration in the bottled water.

The obtained results of the determinations in individual bottled water samples show different tendencies of the compound to release at different temperatures.

In the case of water A, it was observed that the highest concentration occurred at the temperature of 8 °C (11.30 ng/L). At the temperatures of 18 and 28 °C, the amount of bisphenol A in the water remained at a similar level (approx. 10.3 ng/L). The lowest concentration, namely 5.19 ng/L, occurred at the temperature of 38 °C. After the water was exposed to the temperature of 48 °C, an increase in the concentration of bisphenol to the level of 7.84 ng/L was observed.

In the case of water B, for the first two temperatures, i.e., 8 and 18 °C, an increase in the content of bisphenol A from 7.90 ng/L to 10.20 ng/L was noticed, and then, at the temperature of 28 °C, a sharp decrease to the value of 1.63 ng/L (minimum value) occurred. During exposure of the water to the temperatures of 38 and 48 °C, the re-release of bisphenol A into the water was observed in the amounts of 8.00 and 18.54 ng/L, respectively (maximum value).

In the case of water C, the level of bisphenol A at the lowest temperature was equal to approx. 5 ng/L. When the bottle was subjected to the temperatures of 18 and 28 °C, an increase in the amount of migrated bisphenol A was observed, to 9.10 and 10.54 ng/L, respectively. At the temperature of 38 °C, its amount decreased to 6.92 ng/L, and, when the bottle was kept at the temperature of 48 °C, it increased again, reaching the maximum value of approx. 15 ng/L. 

During the series of tests carried out with the water marked as D, an upward trend in the concentration of bisphenol A was initially observed with the temperature increase to 28 °C. At 8 °C, the content was equal to 6.46 ng/L; then, it increased to 13.05 ng/L (18 °C) and finally to 18.03 ng/L (28 °C). At the temperature of 38 °C, there was a sharp decrease in the content of this analyte to the value of 7.00 ng/L, which remained at a similar level at the temperature of 48 °C.

In the case of determinations carried out for water E, for the first two temperatures an increase in the concentration of the compound was observed from 11.50 to 12.45 ng/L (maximum value). At the temperature of 28 °C, it was found that its amount decreased to a value of 9.66 ng/L, and, at the temperature of 38 °C, its concentration further decreased to a value of 6.09 ng/L. At the highest temperature, its content in the water increased again to 10.64 ng/L.

## 4. Discussion

Many people prefer to drink bottled water instead of tap water. Bottled water is sometimes stored in plastic bottles for a long time. These plastic bottles may release some harmful materials into the water, especially when exposed to temperature alteration, which may affect human health. This research work focused on investigating the effect of changing temperature on the quality of bottled water. Based on the analysis, it was found that bisphenol A behaved differently at 38 °C. At this temperature, the water in all bottles had a lower amount of bisphenol A compared to other tested temperatures. The obtained results were below the maximum allowable limit specified in Regulation (EU) No. 213/2018, amounting to 0.05 mg of BPA per kg of food (mg/kg) [36]. The differences between the results obtained in this study and the literature results may be due to the quality and composition of the bottle material, the experimental conditions, and also the storage conditions of the samples before purchase. The conducted experiments show a partial correlation of an increase in the bisphenol A content in water with an increase in the storage temperature of bottled water. An increase in the content of BPA, up to the temperature of 28 °C, is observed; a further increase in temperature does not increase the amount of this compound released into the water from the material. Similar studies have reported the BPA levels in plastic water bottles in multiple countries worldwide [34,37,38,39,40]. BPA is ubiquitous; it is one of the most studied and the most well-known endocrine-disrupting compounds. The literature also clearly indicates that the toxicity of a mixture of endocrine-disrupting compounds is potentially even more harmful than individual exposures to these compounds [41]. Protection of human life, wildlife and Planet Earth require a complete ban of BPA. Therefore, the findings presented in this work are useful for improving the current legislation on bottled waters and their storage.

## 5. Conclusions

Based on the conducted research regarding the assessment of bisphenol A content in bottled waters, it has been confirmed that bisphenol A has the ability to migrate and can penetrate from the packaging of the bottle into the water. This is confirmed by the results of the carried-out tests, in which trace amounts of BPA were detected in each analyzed bottled water exposed to a specific temperature.

The research regarding BPA migration carried out to date has not allowed us to conclude whether a higher temperature results in increased penetration of bisphenol A into the product. It is similar in the case of the presented research because the concentration of bisphenol A was not always the highest at the highest temperature. However, it has undoubtedly been confirmed that temperature is one of the main factors that influences the migration of bisphenol A to products, as was confirmed by determination of the amount of bisphenol A in water without exposing the bottles to different temperatures. 

The levels of toxic BPA determined in this study are below the acceptable limits established by the European Food Safety Commission. The European Commission decided to leave the permissible specific migration limit (SML) for bisphenol A at 0.6 mg/kg (body weight). Despite the fact that the individual concentrations of BPA in bottled water are low (ng/L), the cumulative daily dose in the body may be much higher than the quoted concentrations due to the multitude of daily products containing BPA.

Taking into account the harmful properties of BPA and the potential risk to human health, concentrations of this compound in food should be monitored on an ongoing basis. Product labels, incl. bottled waters, should contain information regarding the composition of its packaging and inform about the possibility of releasing bisphenol A into water.

## Figures and Tables

**Figure 1 ijerph-19-05710-f001:**
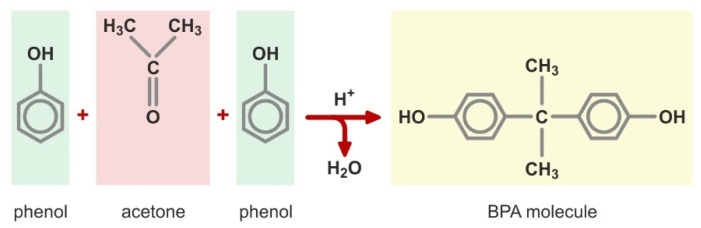
C_15_H_16_O_2_ (R-SO_3_H). The synthesis of bisphenol A.

**Figure 2 ijerph-19-05710-f002:**
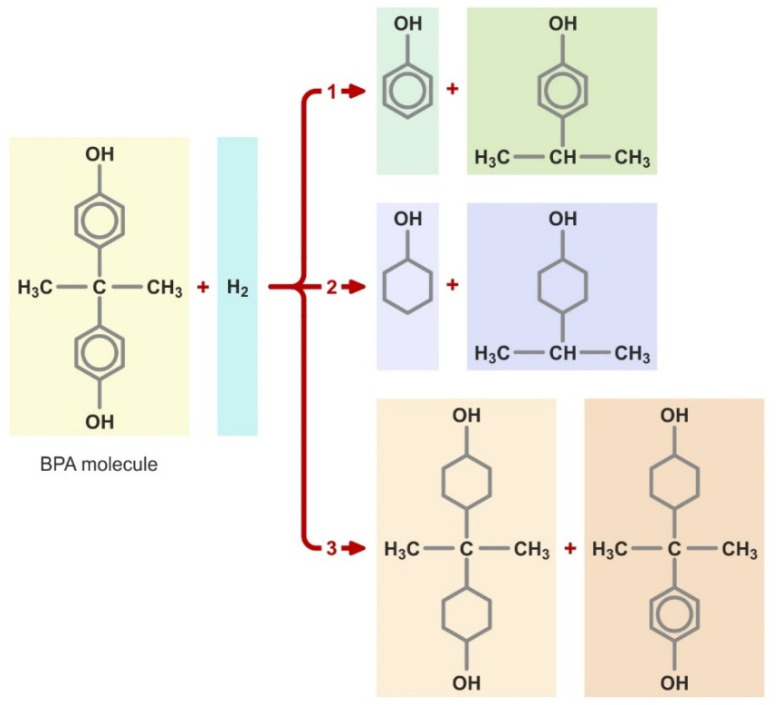
Hydrogenation pathways of BPA.

**Figure 3 ijerph-19-05710-f003:**
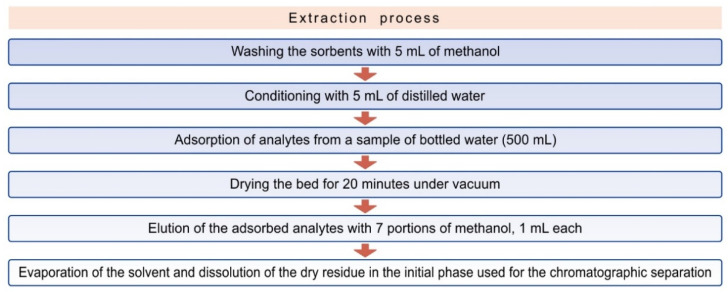
Steps of the SPE extraction process.

**Figure 4 ijerph-19-05710-f004:**
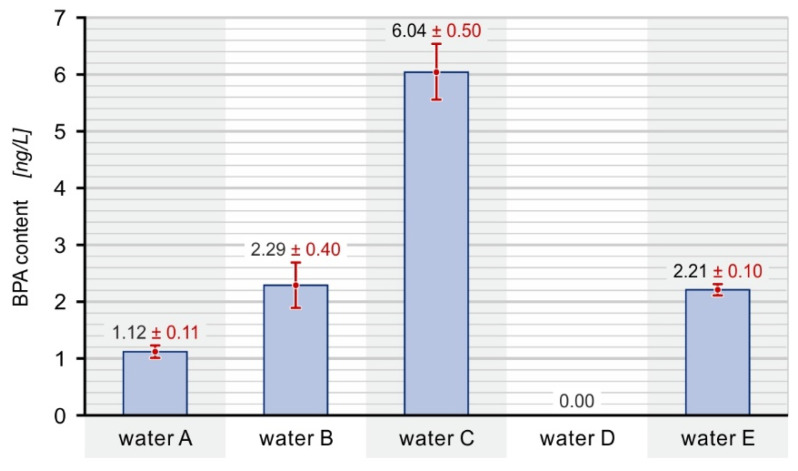
The content of bisphenol A in bottled water immediately after purchase.

**Figure 5 ijerph-19-05710-f005:**
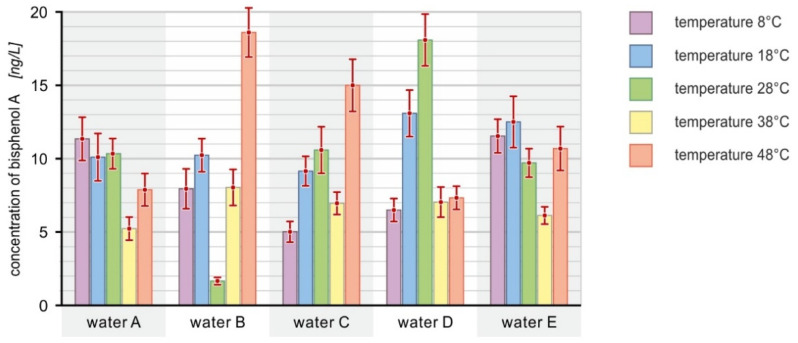
Content of bisphenol A [ng/L] in bottled water depending on the temperature [°C].

**Table 1 ijerph-19-05710-t001:** Optimal MS/MS working conditions in MRM working mode for bisphenol A.

**Compound**	bisphenol A
**Precursor ion [M − H]^−^** [*m*/*z*] **Declustering potential [V]**	227 −75
**MRM 1 transitions-quantitation ion (precursor ion** [*m*/*z*] → **product ion** [*m*/*z*]) **Collision energy (CE) [V]**	227 → 212 −26
**MRM 2 transitions-confirmation ion (precursor ion** [*m*/*z*] → **product ion** [*m*/*z*]) **Collision energy (CE) [V]**	227 → 133 −36

## Data Availability

Not applicable.

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
