# Peer review of "Influence of Temperature on the Quantity of Bisphenol A in Bottled Drinking Water"

_ijerph, 2022, doi:10.3390/ijerph19095710_

Round 1
Reviewer 1 Report
I am glad having an opportunity to review the manuscript entitled "Influence of temperature on the amount of bisphenol A in bottled drinking water".
It is a great paper! I particularly appreciate the main idea of this paper describing the harmful substances found in plastic food packaging. I am not a chemist, but just an ordinary consumer of bottled water (biostatistic), so I cannot criticize the methodology of the chemical research conducted by the authors. An well-written introduction detailing the health risks of bisphenol A.
While reading the submitted manuscript, several questions arose and inaccuracies were also noticed, which I recommend to fix.
- Abstract: The aim and conclusions of the study must be clearly visible. Provide the full names of the abbreviations PET and BPA. Spelling "omount" (L33).
- The purpose of the study should be clearly stated at the end of the introduction, preferably with hypotheses tested. The overall content of the article shows that the focus was on testing the hypothesis that the concentration of bisphenol A in bottled water is temperature dependent. The text in L201-202 is vague and does not reflect the content of the study.
- Material and Methods: L210-211 "A total of 500 mL of bottled water samples A, B, C, D and E, respectively, were subjected to solid phase extraction." Please elaborate on this statement. How many samples were taken from A, B, C, D and E? How many samples were tested at each temperature level?
- A few words are needed about data analysis. In my opinion, the experiment exactly fits the scheme of a two-way factorial analysis (five kinds of bottled water x 5 grades of temperature), isn't it?
- Table 1: Give a short explanation of figures presented.
- Table 2: What were the temperatures of these samples? Why are the concentrations of bisphenol A in these samples much lower than the concentrations given in Figure 3? For example, in sample D, the concentration of bisphenol A was found to be 0.0 (Figure 2), but in Figure 3 it is already high.
- Figure 3: Is it possible to provide p-values or confidence intervals for this data?
- Discussion: The discussion is very limited. Did the authors find similar studies in the literature, not necessarily those of bisphenol A, with which to compare the results of their study? What were the limitations of the study?
- Spelling: L273: 480o; L276: 280o.
Thank you for considering my opinion. I encourage authors to keep on working to improve the manuscript.
Author Response
Answer to Reviewer: 1
We thank the reviewer for their careful reading of the manuscript and their constructive remarks. We have taken the comments on board to improve and clarify the manuscript. Please find below a detailed point-by-point response to all comments. N.B Since the reordering and restructuring of the manuscript was substantial, we have written bullet points of our major changes to the manuscript, and we turned on a ‘track changes’ .
We have substantially changed the text to correct that, especially by being more specific about our objectives, and clearer about the approach adopted. The abstract and the main text in the manuscript were changed to better express the content of the paper, as suggested by Referees.
In the following, we also reply to each of your comments below.
- Abstract: The aim and conclusions of the study must be clearly visible. Provide the full names of the abbreviations PET and BPA. Spelling "omount" (L33).
Answer to point 1.: The abstract has been amended and supplemented, the PET and BPA markings have been clarified.
- The purpose of the study should be clearly stated at the end of the introduction, preferably with hypotheses tested. The overall content of the article shows that the focus was on testing the hypothesis that the concentration of bisphenol A in bottled water is temperature dependent. The text in L201-202 is vague and does not reflect the content of the study.
Answer to point 2.: The main text of the manuscript (especially introduction and text in L201-202) was changed and supplemented with additional informations. The aim of our study stayed clearly stated at the end of the introduction.
- A few words are needed about data analysis. In my opinion, the experiment exactly fits the scheme of a two-way factorial analysis (five kinds of bottled water x 5 grades of temperature), isn't it?
Answer to point 3.: We thank the reviewer for pointing this out, but in our work we wanted only reasearch the problem scale without carrying out a thorough analysis of the obtained results, which we omitted due to the small number of sampled bottled water. Of course, we agree that conducting research on a larger scale will necessitate a broader analysis of the results. Valuable suggestions will be taken into account during the writing the next article.
- Table 1: Give a short explanation of figures presented.
Answer to point 4.: We thank the Reviewer for this remark, but in our opinion the explanation exist in left side of table 1.
- Table 2: What were the temperatures of these samples? Why are the concentrations of bisphenol A in these samples much lower than the concentrations given in Figure 3? For example, in sample D, the concentration of bisphenol A was found to be 0.0 (Figure 2), but in Figure 3 it is already high.
Answer to point 5.: We sincerely thank the Reviewer for this helpful comment to improve the scientific clarity. We have carefully looked through the manuscript and we changed it. Now text is as follows "Among the bottled waters present on the Polish market, five most common brands were selected. In the initial phase of experiment, immediately after their purchase the bisphenol A content was analyzed The concentrations of the bisphenols were de-tected at low levels (at ng/L range). In samples, variations can be observed between samples in consecutive trials. This could be either due to slight differences between bottles because water bottles were purchased from randomly. The results first stage research are presented on figure 4". (L296-302)
- Figure 3: Is it possible to provide p-values or confidence intervals for this data?
Answer to point 6.: We thank the Reviewer for valuable remarks. We have provided errors bars on the figure 3 the revised manuscript.
- Discussion: The discussion is very limited. Did the authors find similar studies in the literature, not necessarily those of bisphenol A, with which to compare the results of their study? What were the limitations of the study?
Answer to point 7.: The discussion was changed and supplemented with additional references.
- Spelling: L273: 480o; L276: 280o
Answer to point 8.: We thank the Reviewer. We improved the errors in the manuscript.
We are grateful for your consideration of this manuscript, and we also very much appreciate your suggestions, which have been very helpful in improving the manuscript. All the comments we received on this study have been taken into account in improving the quality of the article.
We believe that the reviewers’ suggestions have been very helpful in improving this manuscript.
We hope that these changes to the manuscript will facilitate the decision to publish this study. In any case, we are open to consideration of any further comment on our answers.

Reviewer 2 Report
Influence of temperature on the amount of bisphenol A in bottled drinking water by Dobrochna Ginter-Kramarczyk et al. has good scientific merit. Before accepting I have suggestions for authors:
- Abstract: Please improve the writing styles. For example, the starting sentence- the article represents......, mention the significance of the study in one sentence. And the rest of the sentences are also lacks in coherence.
- Introduction: Its too long. I would suggest to write just five paragraphs- introducing the topic, significance of the study, factors affecting the BPA level in general, review and gaps and then finally the aim of the study. I would suggest to avoid any table or figure from the introduction. Please put the points in the line ( line 103).
- Methods: Please use the flow chart to show the steps. Avoid short paragraphs.
- Results: please use the graph for the table 2. Avoid short paras. Please add statistical analyses for showing the significance differences.
- Discussion: Please compare your data with any previous studies.
Author Response
Answer to Reviewer: 2
We thank the reviewer for their careful reading of the manuscript and their constructive remarks. We have taken the comments on board to improve and clarify the manuscript. Please find below a detailed point-by-point response to all comments. N.B Since the reordering and restructuring of the manuscript was substantial, we have written bullet points of our major changes to the manuscript, and we turned on a ‘track changes’ .
We have substantially changed the text to correct that, especially by being more specific about our objectives, and clearer about the approach adopted. The abstract and the main text in the manuscript were changed to better express the content of the paper, as suggested by Referees.
In the following, we also reply to each of your comments below.
1. Abstract: Please improve the writing styles. For example, the starting sentence- the article represents......, mention the significance of the study in one sentence. And the rest of the sentences are also lacks in coherence.
Answer to point 1.: The abstract has been amended and supplemented, the PET and BPA markings have been clarified.
2. Introduction: Its too long. I would suggest to write just five paragraphs- introducing the topic, significance of the study, factors affecting the BPA level in general, review and gaps and then finally the aim of the study. I would suggest to avoid any table or figure from the introduction. Please put the points in the line ( line 103).
Answer to point 2.: We thank the reviewer for the suggestion; however, we believe that for consistence it is better not change form of introduction but we supplemented introduction with additional references
3. Methods: Please use the flow chart to show the steps. Avoid short paragraphs.
Answer to point 3.:We used the flow chart to show the steps (Figure 3. Steps of the extraction process)
4. Results: please use the graph for the table 2. Avoid short paras. Please add statistical analyses for showing the significance differences.
Answer to point 4.:We added the graph (Figure 4. The content of bisphenol A in bottled water immediately after its purchase) instead of a table 2. In addition we completed the chart 5 (Figure 5. Content of bisphenol A [ng / L] in bottled water depending on the temperature [°C]) about the statistical analyses for showing the significance differences.
5. Discussion: Please compare your data with any previous studies.
Answer to point 5.: The discussion was changed and supplemented with additional references.
We are grateful for your consideration of this manuscript, and we also very much appreciate your suggestions, which have been very helpful in improving the manuscript. All the comments we received on this study have been taken into account in improving the quality of the article.
We believe that the reviewers’ suggestions have been very helpful in improving this manuscript.
We hope that these changes to the manuscript will facilitate the decision to publish this study . In any case, we are open to consideration of any further comment on our answers.
Reviewer 3 Report
Ginter-Kramarczyk et al. submitted an interesing article on the influence of temperature on the migration of BPA in bottled drinking water sold in polish market. The topic is extremely current, due to the possible Public Health implications related to exposure to this substance. The article is substantially well structured but needs clarification regarding some methods of conducting the study.
Please take your precious time to assess the following aspects:
L 88: "Studies regarding bottled water have shown that over 90% of the general population is exposed to the harmful effects of BPA" -> please provide specific references, citing them at the end of the sentence
LL 89 - 91: "Recent studies 89 have assessed its content in food products (canned beans, canned vegetables and meat, fresh turkey, infant formulas) and its concentration ranged from 0.23 to 65.0 ng / g per 100 analyzed samples" -> please provide specific references, citing them at the end of the sentence
L 185: “tolerated daily dose” -> it should be Tolerable Daily Intake
L 184: "positive opinion" -> do you mean scientific opinion? The same for L 187
LL 184 - 186: "On this basis, EFSA published a positive opinion and established the tolerated daily dose of TDI at 0.05 mg / kg body weight per day [29]." -> since the sentence refers specifically to an EFSA scientific opinion, I would consider it appropriate to provide a specific reference (citation n. 29 is not congruent).
Materials and Methods section: when was the study conducted? Furthermore, was the plastic making up the food contact material light or dark in color and was it homogeneous by type of brand?
There are some questionable aspects and to be carefully evaluated when Official Controls are carried out at the markets, since you have sampled 25 bottles coming from 5 different brands. The food chain therefore is composed by different manufacturers, which therefore may have had different suppliers, different transport chain (timing and methods), different storage, etc. These variables and co-factors may, in my opinion, have interfered in “not having” a homogeneous aqueous matrix at the initial level. A more appropriate study design should have included the search for BPA at T0 (without heat stress). The interferences may have already occurred before the thermal stress phase conducted in the laboratory, this aspect is to be considered as a possible limitation of the study, which probably should be better explained (compared to what is briefly stated in LL 302 - 305).
Discussions section: I would say that it is appropriate to consider the contents of the following scientific opinion issued by EFSA and to cite it: EFSA CEF Panel (EFSA Panel on Food Contact Materials, Enzymes, Flavourings and Processing Aids), 2015. Scientific Opinion on the risks to public health related to the presence of bisphenol A (BPA) in foodstuffs: Executive summary. EFSA Journal 2015;13(1):3978, 23 pp. doi:10.2903/j.efsa.2015.3978
I also suggest to consider and discuss the very recent hypothesis being formulated by EFSA, concerning the lowering of the TDI limit (https://www.efsa.europa.eu/en/news/bisphenol-efsa-draft-opinion-proposes-lowering -tolerable-daily-intake).
Author Contributions section: please specify in detail how the individual Authors contributed, as required by the Instructions for Authors of IJERPH.
I suggest a careful review since more spaces between one word and another (not necessary - format correctly) are frequently present in the article (especially in the abstract). Finally, the article needs a thorough review in English, both for typos, syntax and form of the English language, perhaps considering having it reread by a native speaker colleague.
Thank you for your efforts in perfecting this article.
Author Response
Answer to Reviewer: 3
We thank the reviewer for their careful reading of the manuscript and their constructive remarks. We have taken the comments on board to improve and clarify the manuscript. Please find below a detailed point-by-point response to all comments. N.B Since the reordering and restructuring of the manuscript was substantial, we have written bullet points of our major changes to the manuscript, and we turned on a ‘track changes’ .
We have substantially changed the text to correct that, especially by being more specific about our objectives, and clearer about the approach adopted. The abstract and the main text in the manuscript were changed to better express the content of the paper, as suggested by Referees.
In the following, we also reply to each of your comments below.
L 88: "Studies regarding bottled water have shown that over 90% of the general population is exposed to the harmful effects of BPA" -> please provide specific references, citing them at the end of the sentence
Answer: The introduction was changed and supplemented with additional references. The unfortunate notation was changed, too.
LL 89 - 91: "Recent studies 89 have assessed its content in food products (canned beans, canned vegetables and meat, fresh turkey, infant formulas) and its concentration ranged from 0.23 to 65.0 ng / g per 100 analyzed samples" -> please provide specific references, citing them at the end of the sentence
Answer: The citation of the literature from which this citation was taken has been added: Schecter A, Malik N, Haffner D, Smith S, Harris TR, Paepke O, Birnbaum L. Bisphenol A (BPA) in U.S. food. Environ Sci Technol. 2010 Dec 15;44(24):9425-30. doi: 10.1021/es102785d. Epub 2010 Nov 1. PMID: 21038926
L 185: “tolerated daily dose” -> it should be Tolerable Daily Intake
L 184: "positive opinion" -> do you mean scientific opinion? The same for L 187
LL 184 - 186: "On this basis, EFSA published a positive opinion and established the tolerated daily dose of TDI at 0.05 mg / kg body weight per day [29]." -> since the sentence refers specifically to an EFSA scientific opinion, I would consider it appropriate to provide a specific reference (citation n. 29 is not congruent).
Answer.: The citation of the literature has been added (LL 184 - 186): https://www.efsa.europa.eu/en/news/bisphenol-efsa-draft-opinion-proposes-lowering-tolerable-daily-intake#efsa-page-title. L 185: change to Tolerable Daily Intake, L 184: change from "positive opinion" to "scientific opinion", L 187: change from "positive opinion" to "scientific opinion"
Materials and Methods section: when was the study conducted? Furthermore, was the plastic making up the food contact material light or dark in color and was it homogeneous by type of brand?
There are some questionable aspects and to be carefully evaluated when Official Controls are carried out at the markets, since you have sampled 25 bottles coming from 5 different brands. The food chain therefore is composed by different manufacturers, which therefore may have had different suppliers, different transport chain (timing and methods), different storage, etc. These variables and co-factors may, in my opinion, have interfered in “not having” a homogeneous aqueous matrix at the initial level. A more appropriate study design should have included the search for BPA at T0 (without heat stress). The interferences may have already occurred before the thermal stress phase conducted in the laboratory, this aspect is to be considered as a possible limitation of the study, which probably should be better explained (compared to what is briefly stated in LL 302 - 305).
Answer: We sincerely thank the Reviewer for this helpful comment to improve the scientific clarity. We have carefully looked through the manuscript and we changed it. Now text is as follows "Water bottles were purchased from randomly selected supermarkets in Poznan. With-in each store, water bottles were also randomly selected from the stocked batches of the product. Production date of these brands was almost the same, and each brand's expiry date was set at one year. The water bottles were of similar shape and volume (500 mL). The plastic material was identified as PET by the global identification number seen inside the recycling triangle on the bottom surface of each bottle. Collec-tively, a total of 25 water bottles were used in this study. Randomly selected water bot-tles were transport to the Science Research Laboratory of Poznan University of Tech-nology and kept in 4 °C until analysis. One series of bottles directly after transported to laboratory was analyzed. The next, water in the bottles was exposed to the tempera-ture of 8, 18, 28, 38 and 48°C for 24 hours. Analysis of each sample for quantitative measurement of BPA was done in five repetitions. The selected temperatures were not random and reflected the conditions that can occur in the temperate climate zone (Poland) at different times of the year". We hope that these changes to the manuscript will facilitate the decision to publish this study. In any case, we are open to consideration of any further comment on our answers.
Discussions section: I would say that it is appropriate to consider the contents of the following scientific opinion issued by EFSA and to cite it: EFSA CEF Panel (EFSA Panel on Food Contact Materials, Enzymes, Flavourings and Processing Aids), 2015. Scientific Opinion on the risks to public health related to the presence of bisphenol A (BPA) in foodstuffs: Executive summary. EFSA Journal 2015;13(1):3978, 23 pp. doi:10.2903/j.efsa.2015.3978
I also suggest to consider and discuss the very recent hypothesis being formulated by EFSA, concerning the lowering of the TDI limit (https://www.efsa.europa.eu/en/news/bisphenol-efsa-draft-opinion-proposes-lowering -tolerable-daily-intake).
Answer: The citation of the literature has been added
Author Contributions section: please specify in detail how the individual Authors contributed, as required by the Instructions for Authors of IJERPH.
Answer: We thank the reviewer for the suggestion, we chacked it.
I suggest a careful review since more spaces between one word and another (not necessary - format correctly) are frequently present in the article (especially in the abstract). Finally, the article needs a thorough review in English, both for typos, syntax and form of the English language, perhaps considering having it reread by a native speaker colleague.
Answer: We thank the reviewer for the suggestion, we chacked it.
Round 2
Reviewer 2 Report
Thank you authors for considering my suggestions.
I think introduction is still too long .
I missed the information about statistical analysis
Author Response
We would like to thank the reviewer again for their careful reading of the manuscript and their constructive remarks.
We have substantially changed the introduction nad we have added an explanation of why the statistical analysis of the results was not performed ("In our work we research only the problem scale without carrying out a thorough sta-tistical analysis of the obtained results, which we omitted due to the small number of sampled bottled water"). This manuscript was edited for proper
English language, grammar, punctuation, spelling, and overall style by qualified native English speaking editor.
Reviewer 3 Report
Thank you for the review and the changes made, the Authors have provided useful clarifications.
I also kindly ask you to consider the following:
LL 91 - 92: "Studies regarding bottled water have shown that of the general population is exposed to the effects of BPA.": unfortunately the sentence is not complete (is something missing?) and has not clarified what was previously requested
L 97: “o1.36 μg / g”: please clarify
I would say that the addition of the following sentences has clarified many aspects related to T0 test: "One series of bottles directly after transported to laboratory was analyzed" and “In the initial phase of experiment, immediately after their purchase the bisphenol A content was analyzed”, and that it was definitely useful to add the Figure n. 4.
Figure 5: there are two images, I believe only one should be kept, probably the one indicating the uncertainty ranges.
Thank you for your efforts to improve this scientific work.
Author Response
We would like to thank the reviewer again for their careful reading of the manuscript and their constructive remarks.
We considered the following sugestions:
LL 91 - 92: "Studies regarding bottled water have shown that of the general population is exposed to the effects of BPA.": unfortunately the sentence is not complete (is something missing?) and has not clarified what was previously requested
Answer: We sincerely thank the Reviewer for this helpful comment to improved the scientific clarity. We removed unfortunate sentence.
L 97: “o1.36 μg / g”: please clarify
Answer: We thank the Reviewer. We improved the errors in the manuscript and we changed the record from from 0.23 to 65.0 ng/g to 0.23 to 65.0 ng/100 g of samples.
Figure 5: there are two images, I believe only one should be kept, probably the one indicating the uncertainty ranges.
Answer: Thank you for sugesstion - we removed the unnecessary chart
This manuscript was edited for proper English language, grammar, punctuation, spelling, and overall style by qualified native English speaking editor.